# A comparative study to optimize experimental conditions of pentylenetetrazol and pilocarpine-induced epilepsy in zebrafish larvae

David Szep[1,2], Bianka Dittrich[1,2], Aniko Gorbe[1,2], Jozsef L. Szentpeteri[1], Nour Aly[2], Meng Jin[3], Ferenc Budan[1,2☯], Attila Sik[1,2,4☯] *

1 Institute of Transdisciplinary Discoveries, Medical School, University of Pecs, Pecs, Hungary, 2 Institute of Physiology, Medical School, University of Pecs, Pecs, Hungary, 3 Biology Institute, Qilu University of Technology (Shandong Academy of Sciences), Ji'nan, Shandong Province, P.R. China, 4 College of Medical and Dental Sciences, University of Birmingham, Birmingham, United Kingdom

☯ These authors contributed equally to this work.
* sik.attila@pte.hu

**Data Availability Statement:** All MS Excel files are available from the EBI database (accession number: S-BSST1111) at https://www.ebi.ac.uk/biostudies/studies/S-BSST1111.

## Abstract

A common way to investigate epilepsy and the effect of antiepileptic pharmaceuticals is to analyze the movement patterns of zebrafish larvae treated with different convulsants like pentylenetetrazol (PTZ), pilocarpine, etc. Many articles have been written on this topic, but the research methods and exact settings are not sufficiently defined in most. Here we designed and executed a series of experiments to optimize and standardize the zebrafish epilepsy model. We found that during the light and the dark trials, the zebrafish larvae moved significantly more in the light, independent of the treatment, both in PTZ and pilocarpine-treated and the control groups. As expected, zebrafish larvae treated with convulsants moved significantly more than the ones in the control group, although this difference was higher between the individuals treated with PTZ than pilocarpine. When examining the optimal observation time, we divided the half-hour period into 5-minute time intervals, and between these, the first 5 minutes were found to be the most different from the others. There were fewer significant differences in the total movement of larvae between the other time intervals. We also performed a linear regression analysis with the cumulative values of the distance moved during the time intervals that fit the straight line. In conclusion, we recommend 30 minutes of drug pretreatment followed by a 10-minute test in light conditions with a 5-minute accommodation time. Our result paves the way toward improved experimental designs using zebrafish to develop novel pharmaceutical approaches to treat epilepsy.

## Introduction

Zebrafish (*Danio rerio*) has widely emerged as a model organism in studies related to neuroscience [1, 2]. Despite their relatively simple nervous system, they show similarities in the development, genetic structure, and function with the mammalian nervous system [3]. The

**Funding:** This work was supported by the Medical Research Council UK (grant number G1001235) and European Union's Horizon 2020 OPEN FET RIA (NEURAM, No, 712821).The funders had no role in study design, data collection and analysis, decision to publish, or preparation of the manuscript.

**Competing interests:** The authors have declared that no competing interests exist.

homology between zebrafish genes and human genes is higher than 70% [4, 5] with many similar functions which makes it an ideal model for drug research and central nervous system (CNS) disorders studies [4, 5]. In addition, a pair of adult fish can produce around 200 eggs a day, which can develop in a matter of hours [4], and larvae can absorb drugs directly from water [6]. The small size of zebrafish larvae makes it perfect for large-scale analysis by fitting one larva per well in a single 96-well plate [7]. In the first week of growth, they show behaviors like escaping, hunting, and negative thigmotaxis by swimming [8]. Also, they can react to visual and acoustic stimuli [9]. As zebrafish larva starts to feed at 5 days post fertilization (dpf) [10] and according to regulations the nonfeeding larva is not considered an animal, thus ethical permit is not required for performing experiments on ≤ 5dpf zebrafish larvae. Hence zebrafish larvae can be considered as a non-animal *in vivo* vertebrate model.

To examine the behavior of zebrafish, automated imaging techniques are frequently used. For instance, Noldus'Ethovision XT software became popular for large-scale imaging and behavioral screening. Such an approach provides precise and effective video monitoring making it easy to track motions, swimming speed, total distance traveled, and other aspects of behaviors [2, 7, 9].

Epilepsy is a CNS disorder where a large number of excitatory neurons fire in synchrony causing behavioral, neurological, and molecular changes [11]. This neurological disorder is due to an imbalance in excitation-inhibition in the CNS [2, 3]. The global prevalence of epilepsy in humans is ca. 1% [12] making this disorder one of the most common. The development of new anti-epileptic drugs (AED) is sought because one-third of patients suffering from epilepsy do not respond to existing AEDs [13].

Pentylenetetrazol (PTZ) is widely used to induce epileptiform activity in zebrafish. It is an antagonist of gamma-aminobutyric acid (GABA) inhibitory neurotransmitter with other multifarious mechanisms of action [3, 14]. The zebrafish larvae are commonly used as an epilepsy model, as they display spontaneous seizures when introduced to PTZ [4]. Different PTZ concentrations trigger locomotor activities in zebrafish larvae in various manners suggesting a non-linear PTZ-dependent fluctuation of anxiety level [15, 16]. Lower concentrations of PTZ can increase locomotion activity and high concentrations of PTZ (15 mM) cause a clonus-like convulsion that can only be reversed by some anti-epileptic drugs [2]. PTZ in 10 mM concentration is ideal for pharmacological tests and the behavioral effects of this concentration can last for approximately 16 hours [10]. The PTZ treatment under different illuminations also alters the anxiety level of zebrafish larvae differently, and the pattern of triggered movement abnormalities has not yet been investigated [3, 16]. On the other hand, pilocarpine, a muscarinic acetylcholine receptor agonist, in 30 mM concentration is another widely used drug in epilepsy research because it induces epileptiform activity that mimics closely the human temporal lobe epilepsy (TLE) [13, 14, 17–20]. The effect of the pilocarpine decreases only slightly in the first 48 hours and persists up to 10 days [21], and spontaneous recurrent seizures can be seen several weeks later [11].

Details of the research methods of epilepsy model of zebrafish are often not sufficiently described [10]. Tracking time of movements and duration time of recording vary in different publications [10]. Only a few studies focused on the methodology have been published, which correlate to specific locomotion detection techniques and drug concentration [3, 10, 16]. For the zebrafish epilepsy model refinement and standardization further experiments are needed [22].

Our research aim was to test the effect of PTZ and pilocarpine on the locomotion of zebrafish larvae in light and dark environments for various recording periods to determine the optimal recording conditions. The purpose of such an optimized method is to support AED tests in the future to develop novel pharmaceutical therapeutic approaches.

## Materials and methods

Adult zebrafish were kept in Tecniplast Zeb Tec System. The temperature of their water was set to 26˚C, the pH to 7.5, and the conductivity to 500 μS. The conductivity level is adjusted by instant sea salt. The system automatically provided a 14-hour-long light period and a 10-hour-long dark period. The fish were fed two times a day. To produce embryos, we used wild-strain zebrafish pairs in breeding tanks. Until the experiment, the larvae were kept in egg water that was changed daily (60μg Red Sea—Coral pro sea salt in 1 ml distilled water) and was incubated at 28˚C [23]. We used a total number of 270 zebrafish larvae (5 dpf), and 90 were used for each experiment. Each experiment was repeated three times. We placed the fish in 6-well plates, filled up with 5 ml of egg water, and divided them equally into three groups (30 in each): control, PTZ and pilocarpine treated. The concentrations we used for the treatment were 10 mM of PTZ [10], and 30 mM of pilocarpine [14, 18] each diluted in distilled water and egg water media for the controls. We left the fish in the solution for 30 minutes for accommodation before starting the recording [10]. We used the Noldus' DanioVision observation chamber (Noldus Information Technology, The Netherlands), and heated the system to 28˚C using the temperature control unit. After the 30-minute accommodation time, 90 fish were moved into a 96-well plate, where each well included one fish and placed the plate into the observation chamber. For recording and data collecting we used Noldus' EthoVisionXT software (Noldus Information Technology, The Netherlands).

We performed three different experiments, and each of them contained three trials: 1) a 10-minute-long dark trial with a tapping stimulus (internal part of the Noldus system, purchased as an option) in the fifth minute, 2) the same length trial with a tapping stimulus, but performed in the light environment, 3) and a 30-minute-long trial in the light without any tapping. Between each trial, there was a 1-minute-long accommodation time, and each trial was repeated three times with each plate (**Fig 1**).

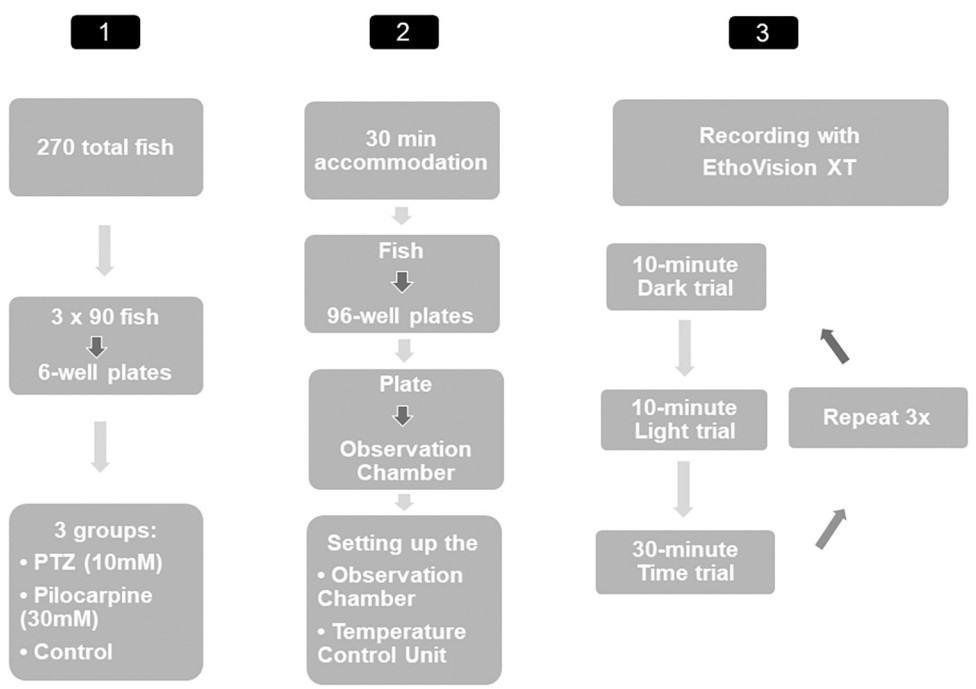

**Fig 1. Flowchart of the research design.**

After the trials to filter out the fish which have not been found by the software ("subjects not found" performance variable), we set up a 0.2% threshold and excluded the remaining data above this limit. After the filtering process for each experiment, we exported the total moved distance data. In the case of the whole period of the dark and the light trials, we summarized the total moved distances of each fish and used these data for further analysis. In the case of the time trials, we cumulated the distances every 5 minutes within the 30-minute-long intervals. These 5-minute periods were compared to each other individually and cumulatively. Before the statistical tests, we used outlier detection with the interquartile range method and filtered out the data as needed. Since our data distributions were not normal according to the Shapiro-Wilk normality test, for statistical analysis we performed Mann-Whitney U tests for the comparison between the groups. The significance level was set to 0.05. We also performed linear regression analysis with the cumulative values of the total moved distance data, to assess the relationship between the variables. For data analysis, we used Microsoft 365 Access, Excel, and Past 4.03 software [23].

## Results

To test the effect of the light and the dark environment, we performed two 10-minute-long trials. We observed that the mean of the total distance moved by the zebrafish larvae was significantly higher ($z = 3.8224$; $p = 0.0003$) in the light trials than in the dark environment (**Fig 2**). Groups of larvae treated with either PTZ or pilocarpine also showed differences between the dark and light trials, thus the difference was independent of the treatment. There was a significant difference between the treatments in every combination (**Table 1**, **Fig 3**). As expected, zebrafish larvae treated with PTZ or pilocarpine moved significantly more than the ones in the control group ($z_{control-PTZ} = 5380$; $p_{control-PTZ} = 8.65E-22$; $z_{control-pilocarpine} = 10473$; $p_{control-pilocarpine} = 6.89E-09$;), although this difference was significantly higher between the individuals treated with PTZ than the ones treated with pilocarpine (**Fig 4**). This difference also turned out to be significant ($z_{PTZ-pilocarpine} = 6574$; $p_{PTZ-pilocarpine} = 5.65E-15$). We also found that the

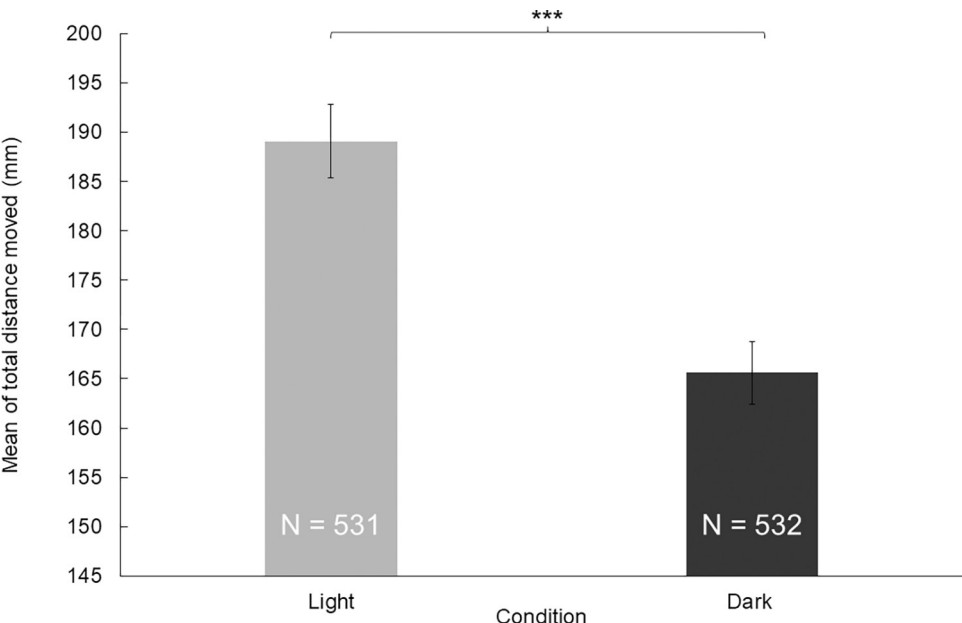

**Fig 2. The difference between the means of moved distances during light and dark trials (Mann-Whitney U test).**

**Table 1. Differences between the treated groups during the light and dark trials (Mann-Whitney U test).**

|  | Light—Control | Light—PTZ | Light—Pilocarpine | Dark—Control | Dark—PTZ | Dark—Pilocarpine |
|---|---|---|---|---|---|---|
| **Light—Control** | - | 6984 | 13763 | 14475 | 9734 | 14834 |
| **Light—PTZ** | 1.18E-15 | - | 6424 | 4908 | 9060.5 | 5056 |
| **Light—Pilocarpine** | 3.35E-05 | 2.63E-18 | - | 10280 | 9784 | 14316 |
| **Dark—Control** | 0.006633 | 1.74E-23 | 5.91E-12 | - | 7070 | 11253 |
| **Dark—PTZ** | 4.64E-10 | 4.20E-05 | 2.98E-10 | 1.01E-17 | - | 7478 |
| **Dark—Pilocarpine** | 0.01137 | 4.00E-23 | 0.001486 | 5.79E-08 | 1.65E-16 | - |

standard errors of the PTZ-treated zebrafish larvae were higher than in the case of the other two groups. Significant differences between the treatments in every combination are found. For significance values see Table 1.

We examined the optimal observation time with a 30-minute-long trial under light conditions. During the data analysis, we divided this period into 5-minute time intervals. Between the groups, the mean of the total distance moved by the PTZ-treated zebrafish larvae differed every time from every other group. However, between the control group, almost every result of the time intervals differed from the pilocarpine-treated zebrafish, except 5 times in the first 5-minutes-long interval. Within the control group, we found that the first 5-minute-long interval differed from every other significantly. On the other hand, except for the first 5 minutes, there were only three significant differences between the 5-to-10-minute interval and the last three intervals. In the case of the PTZ-treated groups'time intervals, there were only three significant differences between the first 5 minutes and the last three 5-minute-long intervals. Within the pilocarpine-treated zebrafish larvae group, the first and the second 5-minute-long interval differed significantly from every other interval and the last 5-minute-long interval, except one time. In conclusion, the first 5 minutes significantly differed from the other time intervals within the groups in all cases except two trials (**Fig 5**). The statistical values of the Mann-Whitney U test can be seen as supporting information (**S1 Fig**). Furthermore, we found significant differences in fewer numbers of cases within each treated group than between the groups (**S1 Fig**). When we analyzed the 5-minute-long intervals by trials, we found that there

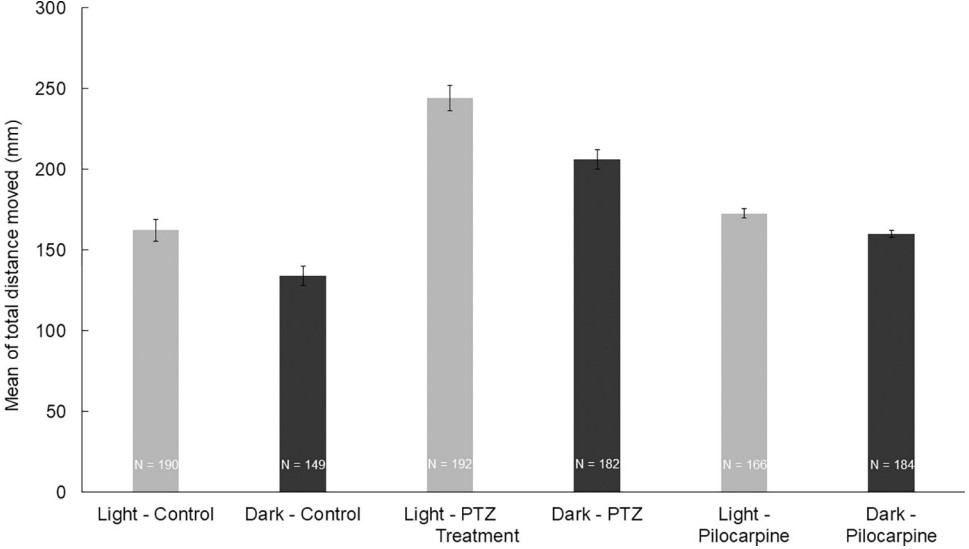

**Fig 3. The means of moved distances between the treated groups during the light and dark trials.**

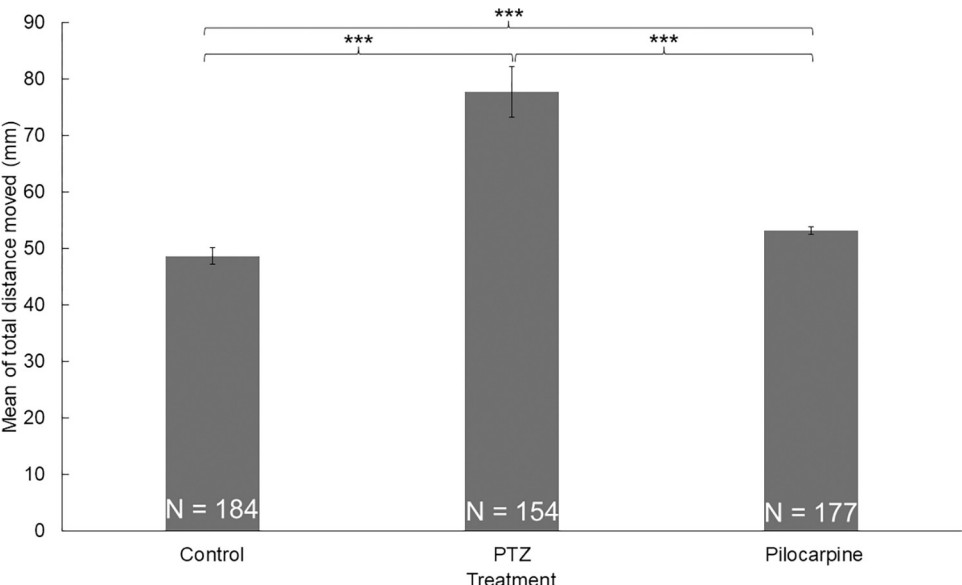

**Fig 4. The difference in the means of total distance moved between the PTZ, pilocarpine treated and control groups (Mann-Whitney U test).**

were more 5 minutes intervals in PTZ-treated groups where the zebrafish showed either hyper-activity or hypoactivity explaining the higher standard errors in this group. On the other hand, more inactive periods were observed in the control group during the first 5 minutes of the recording (**Fig 6**). According to the Mann-Whitney U test the first 5 minutes significantly differed from the other time intervals within the groups in all cases except two trials (also see **S1 Fig**)

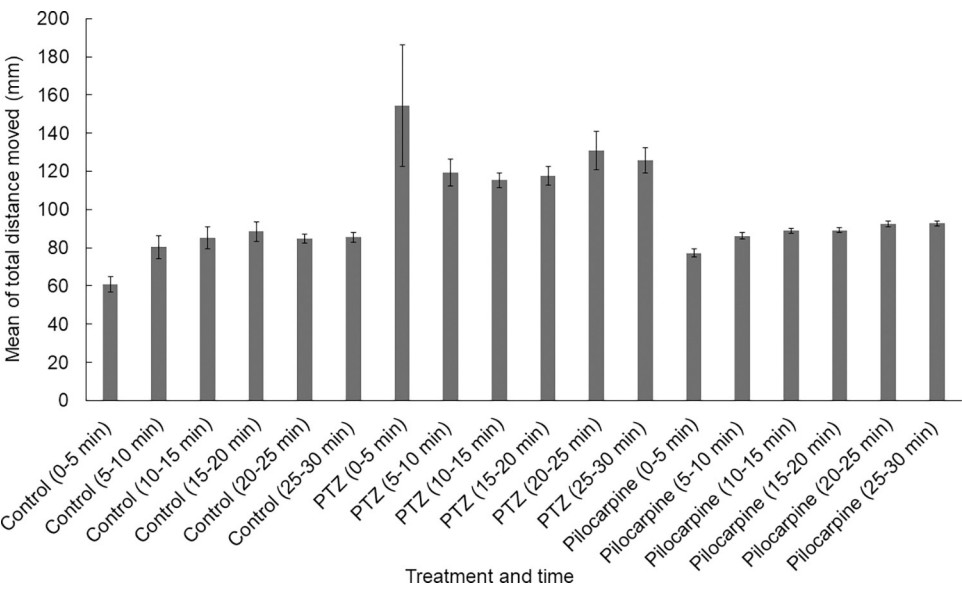

**Fig 5. Means of the total distance moved by the differently treated and control groups in 5-minute time intervals ($N_{control}$ = 184, $N_{PTZ}$ = 154, $N_{pilocarpine}$ = 177).**

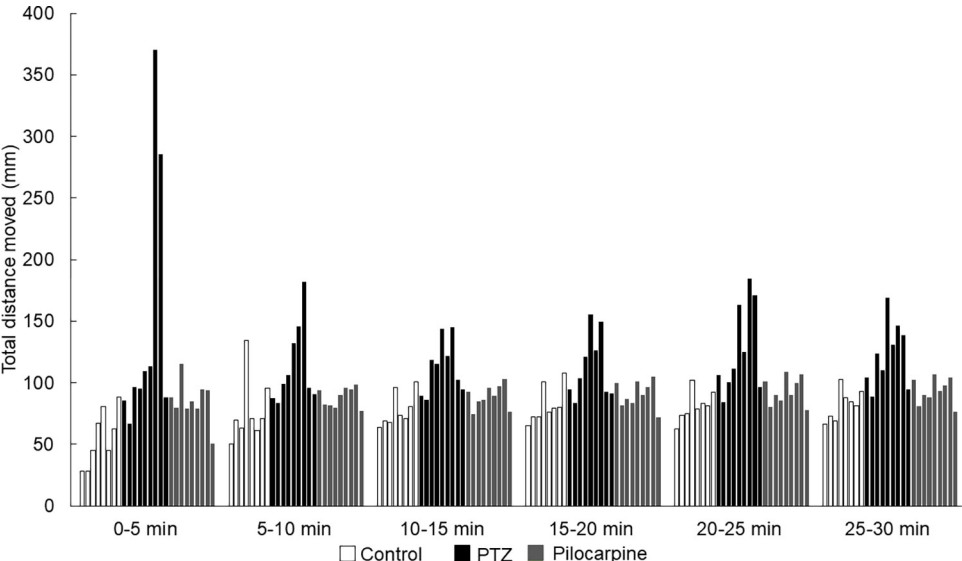

**Fig 6. Mean of total distance moved by the differently treated and control groups during the 9 different trials.**

We also performed a linear regression analysis with the cumulative values of the distance moved during the time intervals to assess if longer observation time could impact the difference between the groups. As seen in **Fig 7** by increasing the time, the distances changed proportionally within the groups. This was supported by the linear regression analysis in most of the cases according to the $R^2$ values (**S1 Table, Fig 8**).

## Discussion

We found that the zebrafish larvae move significantly more in the light than in the dark environment independently of the treatment. Zebrafish have diurnal activity, meaning they are

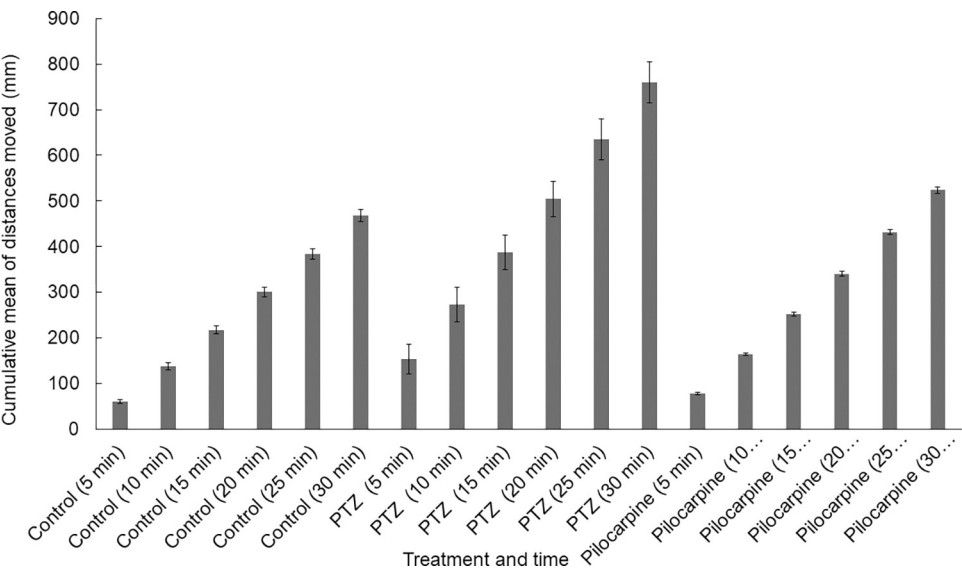

**Fig 7. Cumulative means of distance moved by PTZ, pilocarpine treated and control zebrafish groups.** $N_{control} = 184$, $N_{PTZ} = 154$, $N_{pilocarpine} = 177$.

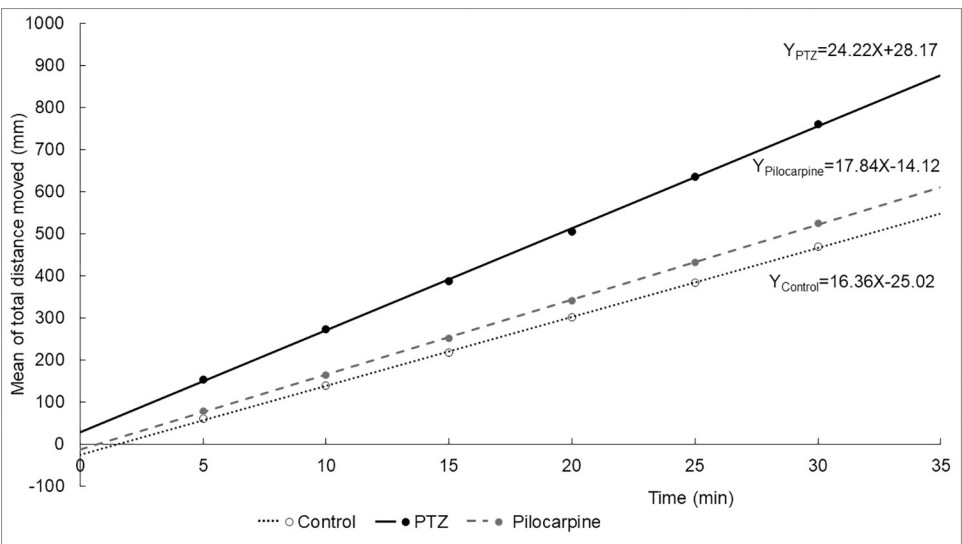

**Fig 8. The regression lines calculated for PTZ, pilocarpine treated and control groups.**

more active during the day and less active at night [24], which might explain their activity under given light conditions. Previous reports described different conditions to test variations in the locomotion of zebrafish larvae [3, 15, 16, 25–27]. The locomotion of the zebrafish differs under different light intensities, depending on the age of the larvae [25]. In contrast to our results, one report showed that the zebrafish moved more in the dark [15]. The reason for the difference in our finding, which is in line with other reports, PTZ increases the sensitivity to changing light environment by activating neurons, which influences the anxiety level of larvae [16]. Other results are in line with our finding: 5 days old zebrafish larvae treated with lower concentrations of PTZ (4 and 8 mM) moved less under dark conditions than under constant light. There was no difference in light and dark activity observed with the use of a higher concentration of PTZ (16 mM) [3], which can be explained by the fact that a high concentration of PTZ, specifically above 10 mM PTZ decreases swimming activity [10]. Under constant light conditions, 5 dpf zebrafish larvae treated with PTZ (8, 16 mM) showed a significant increase in locomotor activity [16]. Pilocarpine enhanced locomotor activity in continuous light. This chemical causes pupil constriction in humans [27] it may cause the same effect in zebrafish [26]. Pilocarpine could make zebrafish larvae less sensitive to light, which is in line with our results because the difference between the light and dark trials was less than the difference between the control and the PTZ-treated group. In the alternating light and dark periods, the zebrafish larvae's movement was higher if the concentration of pilocarpine increased [28]. There was a significant difference between the dark and light trials in the control group, likely due to normal daily activity [24].

The photodegradable active pharmaceutical agents require dark experimental conditions during pharmaceutical tests [29]. The pilocarpine-induced epilepsy model performed in the dark could fulfill this requirement to examine potential photolabile AEDs.

Zebrafish larvae treated with PTZ moved significantly more than the ones treated with pilocarpine, and all of them moved more than the control group. Other authors found that pilocarpine-induced epileptiform activity and seizures were more subtle in comparison to PTZ-induced one [14, 18, 30]. Previous results show that in an acetylcholinesterase inhibitor chlorpyrifos-pretreated experimental model, 1 mM concentration of pilocarpine had a significant startle response relative to control whereas the lower concentration (100 μM) did not [31].

However, there was no significant difference in the response levels between the 100 μM and 1 mM pilocarpine doses [31]. On the other hand, the non-toxic 30 mM concentration of pilocarpine alone is suitable for movement-based pharmacological tests [13, 20].

The PTZ and the pilocarpine mechanism of action are different. PTZ is an antagonist of the GABA(A) receptor increasing the duration of the closed states of the chlorine-ion channels without influencing the conductance or duration of the open state through binding to a specific site partly overlapping with the picrotoxin binding site [32]. On the other hand, pilocarpine activates muscarinic receptor 1 (M1) causing an imbalance of inhibitory and excitatory signal transduction [33]. For example, pilocarpine elevates glutamate levels in the hippocampus after seizures and activates NMDA receptors causing increased excitation in the network [34, 35]. The differences in the mechanism of action can explain the variations we found between PTZ and pilocarpine groups.

We demonstrated that the first 5 minutes of the recording period show the largest difference in the treated groups. According to several publications, zebrafish are supposed to be left to habituate inside the tracking device for 5–10 minutes [4, 16, 30, 36–40]. The habituation is an unlikely explanation for the difference in the first 5 minutes of each recording in our case because after the fish had spent the first three or six trials (40, 80 minutes) in the observation chamber, in the fourth or sixth trial we find that the first 5 minutes of the new recording is different from the other time segments. We suggest that the recording algorithm has some feature that makes the first 5 minutes of recording unreliable, and thus needs to be disregarded.

Other studies used a similar method that we used here (a 30-minute-long trial with 5 minutes intervals), but they observed an increase in the movements of the zebrafish in the first 15 minutes. The likely explanation for this discrepancy is that, unlike other research groups, we used a 30-minute-long accommodation time after the treatment and before starting the recording session [41]. According to Shaw and co-workers [10], other authors used different observation times from 2 minutes to 90 minutes. Vermoesen and colleagues [18] recorded zebrafish locomotor activity for 1 minute only, while the behavior was recorded for 18 minutes by Lopes and co-workers [30] and only 1 and 1.5 minutes were analyzed. Yang, et al. [3] used 10-minute-long trials with 5-minute-long light and dark conditions, while Peng et al. [16] used a 40-minute long light trial and three transient light-dark trials (10-minute light and 5-minute dark) and analyzed only 2-minute-long periods. The measurement of Gawel and colleagues [14] lasted for 18 minutes. Jian et al. [28] observed treated zebrafish with pilocarpine for 20 minutes, which contained a 10-minute-long light, 5-minute-long dark, and 5-minute-long light phases. The effects of the pilocarpine were observed for 8 minutes in 1-minute-long trials, then after a 24- and 96-minute intertrial interval, there were 1-minute-long trials [31]. Based on the linear regression analysis by increasing the observation period, we could see changes in the scale only, not in the differences. Because zebrafish sometimes move erratically to avoid overactive or inactive periods at least 10-minute-long intervals of observation are suggested. The 30-minute-long observation time suggested by Shaw and colleagues [10] is unnecessary. Overall, short observation time can be affected by active and inactive periods of the zebrafish larvae, while the long examination time is unnecessary unless the drug compounds tested at different time intervals provides more valuable insight [4, 39, 42].

## Conclusions

We recommend a 10 minutes observation time instead of 5 minutes, as it filters out the intermittently overactive or inactive periods. If there are no special conditions, we recommend a 30-minute-long drug pretreatment followed by a 15 minutes recording of locomotion in a light environment and discard the first 5 minutes from the subsequent analysis. However, for

the testing of photolabile AED compound candidates, the pilocarpine test in the dark is preferred. Our result paves the way toward improved experimental designs using zebrafish to develop novel pharmaceutical approaches to treat epilepsy. This method is a high-throughput technique, therefore it can be used to test several different combinations of AEDs within a short period.

## Supporting information

**S1 Fig. Differences in the total movement between differently treated zebrafish in 5-minute time intervals (Mann-Whitney U test).**
(TIF)

**S1 Table. Linear regression values of the treated groups during the 9 trials.**
(DOCX)

## Author Contributions

**Conceptualization:** Ferenc Budan, Attila Sik.

**Data curation:** David Szep.

**Formal analysis:** David Szep, Bianka Dittrich, Aniko Gorbe.

**Funding acquisition:** Attila Sik.

**Investigation:** David Szep, Bianka Dittrich, Nour Aly, Meng Jin.

**Supervision:** Attila Sik.

**Validation:** Ferenc Budan.

**Writing – original draft:** David Szep, Bianka Dittrich, Ferenc Budan.

**Writing – review & editing:** Jozsef L. Szentpeteri, Meng Jin, Attila Sik.

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
