## [Decision Letter · Decision Letter 0]

10 May 2023

PONE-D-23-10135A comparative study to optimize experimental conditions of pentylenetetrazol and pilocarpine-induced epilepsy in zebrafish larvaePLOS ONE

Dear Dr. Sik,

Thank you for submitting your manuscript to PLOS ONE. After careful consideration, we feel that it has merit but does not fully meet PLOS ONE’s publication criteria as it currently stands. Therefore, we invite you to submit a revised version of the manuscript that addresses the points raised during the review process. Please submit your revised manuscript by Jun 24 2023 11:59PM. If you will need more time than this to complete your revisions, please reply to this message or contact the journal office at plosone@plos.org. Please include the following items when submitting your revised manuscript:A rebuttal letter that responds to each point raised by the academic editor and reviewer(s). You should upload this letter as a separate file labeled 'Response to Reviewers'.A marked-up copy of your manuscript that highlights changes made to the original version. You should upload this as a separate file labeled 'Revised Manuscript with Track Changes'.An unmarked version of your revised paper without tracked changes. You should upload this as a separate file labeled 'Manuscript'.

We look forward to receiving your revised manuscript.

Kind regards,

Giuseppe Biagini, MD

Academic Editor

PLOS ONE

Reviewers' comments:

Reviewer's Responses to Questions

**Comments to the Author**

1. Is the manuscript technically sound, and do the data support the conclusions?

Reviewer #1: Partly

Reviewer #2: Yes

2. Has the statistical analysis been performed appropriately and rigorously? 

Reviewer #1: Yes

Reviewer #2: I Don't Know

3. Have the authors made all data underlying the findings in their manuscript fully available?

Reviewer #1: Yes

Reviewer #2: Yes

4. Is the manuscript presented in an intelligible fashion and written in standard English?

Reviewer #1: No

Reviewer #2: Yes

5. Review Comments to the Author

Reviewer #1: The zebrafish model is widely used to study the effects of anti-epileptic drugs. However, according to Szep et al, the methods used are often not sufficiently detailed, which leads to results discrepancies between laboratories. In their paper, they performed a series of experiments to optimise the zebrafish epilepsy model and found that the zebrafish larvae move significantly more in the light condition compared to the dark condition, independently of treatment with PTZ or pilocarpine. When examining the optimal observation time, the found that the during the half-hour period, the first 5 minutes were found to be the most different from the others. They therefore recommend 30 minutes of drug pre-treatment followed by a 10-minute test in light conditions with a 5-minute accommodation time. For testing of photolabile AED compound candidates, they recommend the pilocarpine test in the dark. Investigators in the field of epilepsy could benefit from such a refinement of the zebrafish model.

Asbtract

The use of term “epileptics” is not the right term to use. If authors refer to PTZ or pilocarpine, it should be “chemoconvulsants” or “convulsants”.

Materials and Methods

We left the fish in the solution for 30 for accommodation before starting the recording 10. Please correct. 30 s or min?

The Results section should be re-organized. Figure legends should not be placed in the text. Moreover, results of statistical tests are provided in the first paragraph of the Results section but are missing in the following sections.

Discussion

“Zebrafish have diurnal activity, meaning they are more active in the morning and less active at night…”. I think that the term “diurnal” refers to active periods during the day, not only in the morning.

Reviewer #2: The paper entitled “A comparative study to optimize experimental conditions of pentylenetetrazol and pilocarpine-induced epilepsy in zebrafish larvae” by David Szep et.al. describes a series of experiments finalized to the optimization and standardization of the zebrafish larvae epilepsy models of Pentylenetetrazol and pilocarpine to support the development of new anti-epileptic drugs.

The authors describe the steps applied to execute the experiments, how they have analyzed the data and how their findings support their recommendations. The figures are generally well described and easy to understand. Moreover, the recommended method provides an easy readout, the larvae movement, it is a high-throughput technique and a replacement method since it uses not independently feeding larval forms.

Minor issues

Introduction

Line 11: the authors report; “Until 5 dpf larvae is not considered an animal”. According to the Directive 2010/63/EU the law is applicable on the independently feeding larval forms. The housing/rising temperature should also reported since the age at which zebrafish larvea become indipendently feeding forms depends on this parameter.

Materials and methods

The authors should also report in the material and methods:

• the zebrafish strain that they have used to generate the embryos;

• the housing condition of the zebrafish

• the mating system to obtain the zebrafish embryos;

• the incubation/housing conditions of the larvae before the experiments.

Line 3: the authors should also report the recipe of the egg water or quote a publication where is reported;

Line 4: the authors should state in which solvent the drugs are diluted.

Line 6: the authors should add the word “minutes” after “the solution for 30”

Line 9: the authors should write the accommodation time as reported in the Figure 1.

Line 9: “90 fish were moved into a 96 well plate”. The authors should specify if they have placed 30 control, 30 PTZ and 30 pilocarpine treated larvae in each 96 well plate. It is not stated in the manuscript.

Line 19: “we set up a 0.2 % threshold and excluded the remaining data”. The authors should explain what they have done.

Line 25: the authors should describe which method they have used to detect the outlier and what they have considered as an outlier.

Line 26: The authors should consider the use of the Kruskall Wallis test when comparing more than 2 independent groups at the same time (e.g. Fig 4). Mann Whitney U test is suitable for comparing 2 independent groups.

Results

Fig. 4: The authors should state the origin of the data reported in the Figure 4.

Fig. 6: There are 8 white bars instead of 9 bars representing the value for each trial.

Line 34-38 “We also performed a linear regression analysis with the cumulative values of the distance moved during the time intervals to assess if longer observation time could impact the difference between the groups. As seen in Fig 7 by increasing the time, the distances changed proportionally within the groups. This was supported by the linear regression analysis in most of the cases according to the R2 values (S2 Fig).”

The authors to support this statement should assess the possible difference among the regression coefficients (β) of the regression lines calculated for the control, the PTZ and the Pilocarpine groups through a hypothesis test.

Moreover, to support the above statement, the authors should also add a figure showing the regression lines calculated for the control, the PTZ and the Pilocarpine groups.

Discussion

Line 50. “We suggest that the recording algorithm has some feature that makes the first 5 minutes of recording unreliable, and thus needs to be disregarded.”

The author should double check with the recording algorithm maker

6. PLOS authors have the option to publish the peer review history of their article (what does this mean?). If published, this will include your full peer review and any attached files.

Reviewer #1: No

Reviewer #2: No

---

## [Author Response · Author response to Decision Letter 0]

5 Jun 2023

Dear Editors,

We would like to thank the reviewers for their time and very detailed and extremely useful comments on the manuscript. We have corrected the manuscript to address their concerns.

We believe that the manuscript is now suitable for publication in Plos One.

Reviewer #1: 

The zebrafish model is widely used to study the effects of anti-epileptic drugs. However, according to Szep et al, the methods used are often not sufficiently detailed, which leads to results discrepancies between laboratories. In their paper, they performed a series of experiments to optimise the zebrafish epilepsy model and found that the zebrafish larvae move significantly more in the light condition compared to the dark condition, independently of treatment with PTZ or pilocarpine. When examining the optimal observation time, they found that during the half-hour period, the first 5 minutes were found to be the most different from the others. They therefore recommend 30 minutes of drug pre-treatment followed by a 10-minute test in light conditions with a 5-minute accommodation time. For testing photolabile AED compound candidates, they recommend the pilocarpine test in the dark. Investigators in the field of epilepsy could benefit from such a refinement of the zebrafish model.

Abstract

The use of term “epileptics” is not the right term to use. If authors refer to PTZ or pilocarpine, it should be “chemoconvulsants” or “convulsants”.

Thank you for your suggestion, we corrected the revised manuscript and used “convulsant” instead of “epileptics”. 

Materials and Methods

We left the fish in the solution for 30 for accommodation before starting the recording 10. Please correct. 30 s or min? 

Thank you for the comment, it is supposed to be 30 minutes, we corrected it.

The Results section should be re-organized. Figure legends should not be placed in the text.

Thank you for your suggestion, we reorganized as recommended. We removed the figure legends from the text, and placed them to the end, as suggested by the Submission Guidelines.

Moreover, the results of statistical tests are provided in the first paragraph of the Results section but are missing in the following sections.

There would be too many statistical results in the last sections, so in order to read easier the results of the statistical tests are placed into Table 1, S1 FIG, and S1 Table.

Discussion

“Zebrafish have diurnal activity, meaning they are more active in the morning and less active at night…”. I think that the term “diurnal” refers to active periods during the day, not only in the morning. 

We appreciate the comment and corrected the sentence in the following way: “Zebrafish have diurnal activity, meaning they are more active during the day and less active at night.”

Reviewer #2: 

The paper entitled “A comparative study to optimize experimental conditions of pentylenetetrazol and pilocarpine-induced epilepsy in zebrafish larvae” by David Szep et.al. describes a series of experiments finalized to the optimization and standardization of the zebrafish larvae epilepsy models of Pentylenetetrazol and pilocarpine to support the development of new anti-epileptic drugs.

The authors describe the steps applied to execute the experiments, how they have analyzed the data and how their findings support their recommendations. The figures are generally well described and easy to understand. Moreover, the recommended method provides an easy readout, the larvae movement, it is a high-throughput technique and a replacement method since it uses not independently feeding larval forms.

Line 11: the authors report; ““Until 5 dpf larvae is not considered an animal”. According to the Directive 2010/63/EU the law is applicable on the independently feeding larval forms. The housing/rising temperature should also be reported since the age at which zebrafish larvae become independent feeding forms depends on this parameter.

Thank you for your suggestion. The larvae that we used for the experiments did not reach the independent feeding form. To be clear we corrected the sentence in the introduction as the following: “As zebrafish larva starts to feed at 5 days post fertilization (dpf) and according to regulations the nonfeeding larva is not considered an animal, thus ethical permit is not required for performing experiments on ≤ 5dpf zebrafish larvae. Hence zebrafish larvae can be considered as a non-animal in vivo vertebrate model.”

Materials and methods

The authors should also report in the material and methods:

• the zebrafish strain that they have used to generate the embryos; 

• the housing condition of the zebrafish

• the mating system to obtain the zebrafish embryos; 

• the incubation/housing conditions of the larvae before the experiments. 

Line 3: the authors should also report the recipe of the egg water or quote a publication where is reported; 

Line 4: the authors should state in which solvent the drugs are diluted.

Line 6: the authors should add the word “minutes” after “the solution for 30”

Line 9: the authors should write the accommodation time as reported in the Figure 1.

Line 9: “90 fish were moved into a 96 well plate”. The authors should specify if they have placed 30 control, 30 PTZ and 30 pilocarpine treated larvae in each 96 well plate. It is not stated in the manuscript.

Line 19: “we set up a 0.2 % threshold and excluded the remaining data”. The authors should explain what they have done.

Line 25: the authors should describe which method they have used to detect the outlier and what they have considered as an outlier.

We extended the Material and methods part with the missing information.

Line 26: The authors should consider the use of the Kruskall Wallis test when comparing more than 2 independent groups at the same time (e.g. Fig 4). Mann Whitney U test is suitable for comparing 2 independent groups.

We indeed performed Kruskall-Wallis tests during the initial analysis which resulted in very similar results to Mann-Whitney U test. K-W test shows us that there is at least one group that is different from the others, but we were more interested in which one is different. We had the option to use Dunn’s post hoc test along with the K-W test or the M-W U test alone. We chose the M-W U test because in it is more straightforward and gave identical statistical significance to K-W.

Results

Fig. 4: The authors should state the origin of the data reported in the Figure 4.

To create the figures, we used our experimental data. In Figure 4, the significant differences were the result of the Mann-Whitney U test. We extended the figure legends with the name of the test we used.

Fig. 6: There are 8 white bars instead of 9 bars representing the value for each trial.

Thank you for your observation, one row of data was missing, we corrected it in the revised manuscript.

Line 34-38: “We also performed a linear regression analysis with the cumulative values of the distance moved during the time intervals to assess if longer observation time could impact the difference between the groups. As seen in Fig 7 by increasing the time, the distances changed proportionally within the groups. This was supported by the linear regression analysis in most of the cases according to the R2 values (S2 Fig).”

The authors to support this statement should assess the possible difference among the regression coefficients (β) of the regression lines calculated for the control, the PTZ and the Pilocarpine groups through a hypothesis test.

Moreover, to support the above statement, the authors should also add a figure showing the regression lines calculated for the control, the PTZ and the Pilocarpine groups.

Thank you for your suggestion, we created a new figure (Figure 8) showing the regression lines, among the missing regression coefficients (β) values.

Discussion

Line 50. “We suggest that the recording algorithm has some feature that makes the first 5 minutes of recording unreliable, and thus needs to be disregarded.”

The author should double check with the recording algorithm maker.

We indeed contacted the software developer company when we first observed this peculiar problem. After several lengthy discussions, they were unable to identify the reason for this reproducible observation. Since we are not in a position to find the real cause, we left the text in the manuscript as an observation and recommendation to avoid errors in the analysis without an explanation for the observation. 

Finally, we would like to thank the reviewers and editors for evaluating our manuscript. We corrected all errors and followed the reviewers’ suggestions which certainly greatly improved the quality of the manuscript.

---

## [Decision Letter · Decision Letter 1]

6 Jul 2023

A comparative study to optimize experimental conditions of pentylenetetrazol and pilocarpine-induced epilepsy in zebrafish larvae

PONE-D-23-10135R1

Dear Dr. Sik,

We’re pleased to inform you that your manuscript has been judged scientifically suitable for publication and will be formally accepted for publication once it meets all outstanding technical requirements.

Kind regards,

Giuseppe Biagini, MD

Academic Editor

PLOS ONE

Additional Editor Comments (optional):

Reviewers' comments:

Reviewer's Responses to Questions

**Comments to the Author**

1. If the authors have adequately addressed your comments raised in a previous round of review and you feel that this manuscript is now acceptable for publication, you may indicate that here to bypass the “Comments to the Author” section, enter your conflict of interest statement in the “Confidential to Editor” section, and submit your "Accept" recommendation.

Reviewer #1: All comments have been addressed

Reviewer #2: All comments have been addressed

2. Is the manuscript technically sound, and do the data support the conclusions?

Reviewer #1: Yes

Reviewer #2: (No Response)

3. Has the statistical analysis been performed appropriately and rigorously? 

Reviewer #1: Yes

Reviewer #2: (No Response)

4. Have the authors made all data underlying the findings in their manuscript fully available?

Reviewer #1: Yes

Reviewer #2: (No Response)

5. Is the manuscript presented in an intelligible fashion and written in standard English?

Reviewer #1: Yes

Reviewer #2: (No Response)

6. Review Comments to the Author

Reviewer #1: (No Response)

Reviewer #2: (No Response)

7. PLOS authors have the option to publish the peer review history of their article (what does this mean?). If published, this will include your full peer review and any attached files.

Reviewer #1: No

Reviewer #2: No

---

## [Editor Report · Acceptance letter]

20 Jul 2023

PONE-D-23-10135R1 

A comparative study to optimize experimental conditions of pentylenetetrazol and pilocarpine-induced epilepsy in zebrafish larvae 

Dear Dr. Sik:

I'm pleased to inform you that your manuscript has been deemed suitable for publication in PLOS ONE. Congratulations! Your manuscript is now with our production department. 

Kind regards, 

on behalf of

Dr. Giuseppe Biagini 

Academic Editor

PLOS ONE